# Greedy, Joint Syntactic-Semantic Parsing with Stack LSTMs

## Abstract

We present a transition-based parser that jointly produces syntactic and semantic dependencies. It learns a representation of the entire algorithm state, using stack long short-term memories. Our greedy inference algorithm has linear time, including feature extraction. On the CoNLL 2008–9 English shared tasks, we obtain the best parsing performance among models that jointly learn syntax and semantics.

## 1 Introduction

We introduce a new joint syntactic and semantic dependency parser. Our parser draws from the algorithmic insights of the incremental structure building approach of Henderson et al. (2008), with two key differences. First, it learns representations for the parser's entire algorithmic state, not just the top items on the stack or the most recent parser states; in fact, it uses no expert-crafted features at all. Second, it uses entirely greedy inference rather than beam search. We find that it outperforms all previous joint parsing models, including Henderson et al. (2008) and variants (Gesmundo et al., 2009; Titov et al., 2009; Henderson et al., 2013) on the CoNLL 2008 and 2009 (English) shared tasks. Our multilingual results are comparable to the top systems at CoNLL 2009.

Joint models like ours have frequently been proposed as a way to avoid cascading errors in NLP pipelines; varying degrees of success have been attained for a range of joint syntactic-semantic analysis tasks (Sutton and McCallum, 2005; Henderson et al., 2008; Toutanova et al., 2008; Johansson, 2009; Lluís et al., 2013, *inter alia*).

One reason pipelines often dominate is that they make available the complete syntactic parse tree, and arbitrarily-scoped syntactic features—such as the "path" between predicate and argument, proposed by Gildea and Jurafsky (2002)—for semantic analysis. Such features are a mainstay of high-performance semantic role labeling (SRL) systems (Roth and Woodsend, 2014; Lei et al., 2015; FitzGerald et al., 2015; Foland and Martin, 2015), but they are expensive to extract (Johansson, 2009; He et al., 2013).

This study shows how recent advances in representation learning can bypass those expensive features, discovering cheap alternatives available during a greedy parsing procedure. The specific advance we employ is the stack LSTM (Dyer et al., 2015), a neural network that continuously summarizes the contents of the stack data structures in which a transition-based parser's state is conventionally encoded. Stack LSTMs were shown to obviate many features used in syntactic dependency parsing; here we find them to do the same for joint syntactic-semantic dependency parsing.

We believe this is an especially important finding for *greedy* models that cast parsing as a sequence of decisions made based on algorithmic state, where linguistic theory and researcher intuitions offer less guidance in feature design.

Our system's performance does not match that of the top expert-crafted feature-based systems (Zhao et al., 2009; Björkelund et al., 2010; Roth and Woodsend, 2014; Lei et al., 2015), systems which perform optimal decoding (Täckström et al., 2015), or of systems that exploit additional, differently-annotated datasets (FitzGerald et al., 2015). Many advances in those systems are orthogonal to our model, and we expect future work to achieve further gains by integrating them.

Because our system is very fast—with an end-to-end runtime of 177.6±18 seconds to parse the

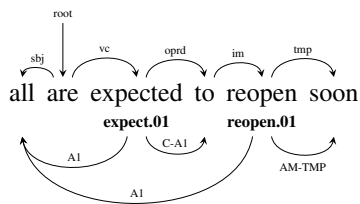

Figure 1: Example of a joint parse. Syntactic dependencies are shown by arcs above the sentence and semantic dependencies, below; predicates are marked in boldface. Correspondences between dependencies might be close (between *expected* and *to*) or not (between *reopen* and *all*).

CoNLL 2009 English test data on a single core—we believe it will be useful in practical settings. An open-source implementation will be made available on publication.

## 2 Joint Syntactic and Semantic Dependency Parsing

We largely follow the transition-based, synchronized algorithm of Henderson et al. (2013) to predict joint parse structures. The input to the algorithm is a sentence annotated with part-of-speech tags. The output consists of a labeled syntactic dependency tree and a directed SRL graph, in which a subset of words in the sentence are selected as predicates, disambiguated to a sense, and linked by labeled, directed edges to their semantic arguments and modifiers. Figure 1 shows an example.

### 2.1 Transition-Based Procedure

The two parses are constructed in a bottom-up fashion, incrementally processing words in the sentence from left to right. The state of the parsing algorithm at timestep $t$ is represented by three stack data structures: a syntactic stack $S_t$, a semantic stack $M_t$—each containing partially built structures—and a buffer of input words $B_t$. Our algorithm also places partial syntactic and semantic parse structures onto the front of the buffer, so it is also implemented as a stack. Each arc in the output corresponds to a transition (or "action") chosen based on the current state; every transition modifies the state by updating $S_t$, $M_t$, and $B_t$ to $S_{t+1}$, $M_{t+1}$, and $B_{t+1}$, respectively. While each state may license several valid actions, each action has a deterministic effect on the state of the algorithm.

Initially, $S_0$ and $M_0$ are empty, and $B_0$ contains the input sentence with the first word at the front of $B$ and a special root symbol at the end.[1] Execution ends on iteration $t$ such that $B_t$ is empty and $S_t$ and $M_t$ contain only a single structure headed by root.

### 2.2 Transitions for Joint Parsing

There are separate sets of syntactic and semantic transitions; the former manipulate $S$ and $B$, the latter $M$ and $B$. All are formally defined in Table 1. The syntactic transitions are from the "arc-eager" algorithm of Nivre (2008). They include:

- S-SHIFT, which copies[2] an item from the front of $B$ and pushes it on $S$.
- S-REDUCE pops an item from $S$.
- S-RIGHT($\ell$) creates a syntactic dependency. Let $u$ be the element at the top of $S$ and $v$ be the element at the front of $B$. The new dependency has $u$ as head, $v$ as dependent, and label $\ell$. $u$ is popped off $S$, and the resulting structure, rooted at $u$, is pushed on $S$. Finally, $v$ is copied to the top of $S$.
- S-LEFT($\ell$) creates a syntactic dependency with label $\ell$ in the reverse direction as S-RIGHT. The top of $S$, $u$ is popped. The front of $B$, $v$ is replaced by the new structure, rooted at $v$.

The semantic transitions are similar, operating on the semantic stack.

- M-SHIFT removes an item from the front of $B$ and pushes it on $M$.
- M-REDUCE pops an item from $M$.
- M-RIGHT($r$) creates a semantic dependency. Let $u$ be the element at the top of $M$ and $v$, the front of $B$. The new dependency has $u$ as head, $v$ as dependent, and label $r$. $u$ is popped off the semantic stack, and the resulting structure, rooted at $u$, is pushed on $M$.
- M-LEFT($r$) creates a semantic dependency with label $r$ in the reverse direction as M-RIGHT. The buffer front, $v$ is replaced by the new $v$-rooted structure. $M$ remains unchanged.

Because SRL graphs allow a node to be a semantic argument of two parents—like *all* in the

---

[1] This works better for the arc-eager algorithm (Ballesteros and Nivre, 2013), in contrast to Henderson et al. (2013), who initialized with root at the buffer front.

[2] Note that in the original arc-eager algorithm (Nivre, 2008), SHIFT and RIGHT-ARC actions move the item on the buffer front to the stack, whereas we only copy it (to allow the semantic operations to have access to it).

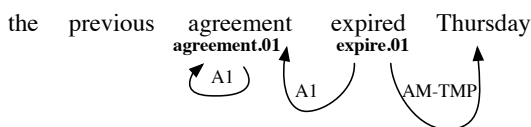

the previous agreement expired Thursday

Figure 2: Example of an SRL graph with an arc from predicate **agreement.01** to itself, filling the A1 role. Our SELF(A1) transition allows recovering this semantic dependency.

example in Figure 1—M-LEFT and M-RIGHT do not remove the dependent from the semantic stack and buffer respectively, unlike their syntactic equivalents, S-LEFT and S-RIGHT. We use two other semantic transitions from Henderson et al. (2013) which have no syntactic analogues:

- M-SWAP swaps the top two items on $M$, to allow for crossing semantic arcs.
- M-PRED($p$) marks the item at the front of $B$ as a semantic predicate with the sense $p$, and replaces it with the disambiguated predicate.

The CoNLL 2009 corpus introduces semantic self-dependencies where many nominal predicates (from NomBank) are marked as their own arguments; these account for 6.68% of all semantic arcs in the English corpus. An example involving an eventive noun is shown in Figure 2. We introduce a new semantic transition to handle such cases:

- M-SELF($r$) adds a dependency, with label $r$ between the item at the front of $B$ and itself. The result replaces the item at the front of $B$.

Note that the syntactic and semantic transitions both operate on the same buffer, though they independently specify the syntax and semantics, respectively. In order to ensure that both syntactic and semantic parses are produced, the syntactic and semantic transitions are interleaved. Only syntactic transitions are considered until a transition is chosen that copies an item from the buffer front to the syntactic stack (either S-SHIFT or S-RIGHT). The algorithm then switches to semantic transitions until a buffer-modifying transition is taken (M-SHIFT).[3] At this point, the buffer is modified and the algorithm returns to syntactic transitions. This implies that, for each word, its left-side

---

[3]Had we *moved* the item at the buffer front during the syntactic transitions, it would have been unavailable for the semantic transitions, hence we only *copy* it.

syntactic dependencies are resolved before its left-side semantic dependencies. For interested readers, an example run of the algorithm is given in §A.

### 2.3 Constraints on Transitions

To ensure that the parser never enters an invalid state, the sequence of transitions is constrained, following Henderson et al. (2013). Actions that copy or move items from the buffer (S-SHIFT, S-RIGHT and M-SHIFT) are forbidden when the buffer is empty. Actions that pop from a stack (S-REDUCE and M-REDUCE) are forbidden when that stack is empty. We disallow actions corresponding to the same dependency, or the same predicate to be repeated in the sequence. Repetitive M-SWAP transitions are disallowed to avoid infinite swapping. Finally, as noted above, we restrict the parser to syntactic actions until it needs to shift an item from $B$ to $S$, after which it can only execute semantic actions until it executes an M-SHIFT.

Asymptotic runtime complexity of this greedy algorithm is linear in the length of the input, as analyzed by Nivre (2009).

## 3 Statistical Model

The transitions in §2 describe the execution paths our algorithm can take; like past work, we apply a statistical classifier to decide which transition to take at each timestep, given the current state. (A full example of a transition sequence is given in the supplementary material.) The novelty of our model is that it learns a finite-length vector representation of the entire joint parser's state ($S$, $M$, and $B$) in order to make this decision.

### 3.1 Stack Long Short-Term Memory (LSTM)

LSTMs are recurrent neural networks equipped with specialized memory components in addition to a hidden state (Hochreiter and Schmidhuber, 1997; Graves, 2013) to model sequences. Stack LSTMs (Dyer et al., 2015) are LSTMs that allow for stack-based operations: *query*, *push*, and *pop*. A "stack pointer" is maintained which determines which cell in the LSTM provides memory unit and the hidden unit when computing the new memory cell contents. *Query* provides a summary of the stack in a single fixed-length vector. *Push* adds an element to the top of the stack, resulting in a new summary. *Pop*, which does not correspond to

| $S_t$ | $M_t$ | $B_t$ | Action | $S_{t+1}$ | $M_{t+1}$ | $B_{t+1}$ | Dependency |
|---|---|---|---|---|---|---|---|
| $S$ | $M$ | $(\mathbf{v},v),B$ | S-Shift | $(\mathbf{v},v),S$ | $M$ | $(\mathbf{v},v),B$ | — |
| $(\mathbf{u},u),S$ | $M$ | $B$ | S-Reduce | $S$ | $M$ | $B$ | — |
| $(\mathbf{u},u),S$ | $M$ | $(\mathbf{v},v),B$ | S-Right($\ell$) | $(\mathbf{v},v),(g_s(\mathbf{u},\mathbf{v},\mathbf{l}),u),S$ | $M$ | $(\mathbf{v},v),B$ | $\mathcal{S}\cup u \xrightarrow{\ell} v$ |
| $(\mathbf{u},u),S$ | $M$ | $(\mathbf{v},v),B$ | S-Left($\ell$) | $S$ | $M$ | $(g_s(\mathbf{v},\mathbf{u},\mathbf{l}),v),B$ | $\mathcal{S}\cup u \xleftarrow{\ell} v$ |
| $S$ | $M$ | $(\mathbf{v},v),B$ | M-Shift | $S$ | $(\mathbf{v},v),M$ | $B$ | — |
| $S$ | $(\mathbf{u},u),M$ | $B$ | M-Reduce | $S$ | $M$ | $B$ | — |
| $S$ | $(\mathbf{u},u),M$ | $(\mathbf{v},v),B$ | M-Right($r$) | $S$ | $(g_m(\mathbf{u},\mathbf{v},\mathbf{r}),u),M$ | $(\mathbf{v},v),B$ | $\mathcal{M}\cup u \xrightarrow{r} v$ |
| $S$ | $(\mathbf{u},u),M$ | $(\mathbf{v},v),B$ | M-Left($r$) | $S$ | $(\mathbf{u},u),M$ | $(g_m(\mathbf{v},\mathbf{u},\mathbf{r}),v),B$ | $\mathcal{M}\cup u \xleftarrow{r} v$ |
| $S$ | $(\mathbf{u},u),(\mathbf{v},v),M$ | $B$ | M-Swap | $S$ | $(\mathbf{v},v),(\mathbf{u},u),M$ | $B$ | — |
| $S$ | $M$ | $(\mathbf{v},v),B$ | M-Pred($p$) | $S$ | $M$ | $(g_d(\mathbf{v},\mathbf{p}),v),B$ | — |
| $S$ | $M$ | $(\mathbf{v},v),B$ | M-Self($r$) | $S$ | $M$ | $(g_m(\mathbf{v},\mathbf{v},\mathbf{r}),v),B$ | $\mathcal{M}\cup v \xleftrightarrow{r} v$ |

Table 1: Parser transitions along with the modifications to the stacks and the buffer resulting from each. Syntactic transitions are shown above, semantic below. Script symbols denote symbolic representations of words and relations, and bold symbols indicate (learned) embeddings (§3.5) of words and relations; each element in a stack or buffer includes both symbolic and vector representations, either atomic or recursive. $\mathcal{S}$ represents the set of syntactic transitions; $\mathcal{M}$ the set of semantic transitions.

a conventional LSTM operation, moves the stack pointer to the preceding timestep, resulting in a stack summary as it was before the popped item was observed. Implementation details (Dyer et al., 2015; Goldberg, 2015) and code were made publicly available.[4]

Using stack LSTMs, we construct a representation of the algorithm state by decomposing it into smaller pieces that are combined by recursive function evaluations (similar to the way a list is built by a *concatenate* operation that operates on a list and an element). This enables information that would be distant from the "top" of the stack to be carried forward, potentially helping the learner.

### 3.2 Stack LSTMs for Joint Parsing

Our algorithm employs four stack LSTMs, one each for the $S$, $M$, and $B$ data structures, so that already-built partial structures are available to the classifier. Like Dyer et al. (2015), we use a fourth stack LSTM, $A$, for the history of actions—$A$ is never popped from, only pushed to. Figure 3 illustrates the architecture. The algorithm's state at timestep $t$ is encoded by the four vectors summarizing the four stack LSTMs, and this is the input to the classifier that chooses among the allowable transitions at that timestep.

Let $\mathbf{s}_t$, $\mathbf{m}_t$, $\mathbf{b}_t$, and $\mathbf{a}_t$ denote the summaries of $S_t$, $M_t$, $B_t$, and $A_t$, respectively. Let $\mathcal{A}_t = \text{Allowed}(S_t, M_t, B_t, A_t)$ denote the allowed transitions given the current stacks and buffer. The parser state at time $t$ is given by a rectified linear unit (Nair and Hinton, 2010) in vector $\mathbf{y}_t$:

$$\mathbf{y}_t = \text{elementwisemax}\left\{\mathbf{0}, \mathbf{d} + \mathbf{W}[\mathbf{s}_t; \mathbf{m}_t; \mathbf{b}_t; \mathbf{a}_t]\right\}$$

---

[4]https://github.com/clab/lstm-parser

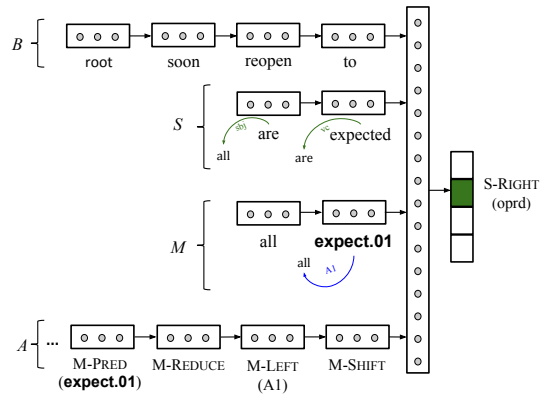

Figure 3: Stack LSTM for joint parsing. The state illustrated corresponds to the ***-marked row in the example transition sequence in Fig. 5 in the supplementary material.

where $\mathbf{W}$ and $\mathbf{d}$ are the parameters of the classifier. The transition selected at timestep $t$ is

$$\arg\max_{\tau \in \mathcal{A}_t} q_\tau + \boldsymbol{\theta}_\tau \cdot \mathbf{y}_t \qquad (1)$$

$$\equiv \arg\max_{\tau \in \mathcal{A}_t} \text{score}(\tau; S_t, M_t, B_t, A_t) \qquad (2)$$

where $\boldsymbol{\theta}_\tau$ and $q_\tau$ are parameters for each transition type $\tau$. Note that only allowed transitions are considered in the decision rule (see §2.3).

### 3.3 Composition Functions

To use stack LSTMs, we require vector representations of the elements that are stored in the stacks. Specifically, we require vector representations of atoms (words, possibly with part-of-speech tags) and parse fragments. Word vectors can be pre-trained or learned directly; we consider a concate-

nation of both in our experiments; part-of-speech vectors are learned and concatenated to the same.

To obtain vector representations of parse fragments, we use neural networks which recursively compute representations of the complex structured output (Dyer et al., 2015). The tree structures here are always ternary trees, with each internal node's three children including a head, a dependent, and a label. The vectors for leaves are word vectors and vectors corresponding to syntactic and semantic relation types.

The vector for an internal node is a squashed ($\tanh$) affine transformation of its children's vectors. For syntactic and semantic attachments, respectively, the composition function is:

$$g_s(\mathbf{v}, \mathbf{u}, \mathbf{l}) = \tanh(\mathbf{Z}_s[\mathbf{v}; \mathbf{u}; \mathbf{l}] + \mathbf{e}_s) \qquad (3)$$

$$g_m(\mathbf{v}, \mathbf{u}, \mathbf{r}) = \tanh(\mathbf{Z}_m[\mathbf{v}; \mathbf{u}; \mathbf{r}] + \mathbf{e}_m) \qquad (4)$$

where $\mathbf{v}$ and $\mathbf{u}$ are vectors correspodning to atomic words or composed parse fragments; $\mathbf{l}$ and $\mathbf{r}$ are learned vector representations for a syntactic and semantic labels respectively. Syntactic and semantic parameters are separated ($\mathbf{Z}_s$, $\mathbf{e}_s$ and $\mathbf{Z}_m$, $\mathbf{e}_m$, respectively).

Finally, for predicates, we use another recursive function to compose the word representation, $\mathbf{v}$ with a learned representation for the dismabiguated sense of the predicate, $\mathbf{p}$:

$$g_d(\mathbf{v}, \mathbf{p}) = \tanh(\mathbf{Z}_d[\mathbf{v}; \mathbf{p}] + \mathbf{e}_d) \qquad (5)$$

where $\mathbf{Z}_d$ and $\mathbf{e}_d$ are parameters of the model. Note that, because syntactic and semantic transitions are interleaved, the fragmented structures are a blend of syntactic and semantic compositions. Figure 4 shows an example.

### 3.4 Training

Training the classifier requires transforming each training instance (a joint parse) into a transition sequence, a deterministic operation under our transition set. Given a collection of algorithm states at time $t$ and correct classification decisions $\tau_t$, we minimize the sum of log-loss terms, given (for one timestep) by:

$$-\log \frac{\exp(q_{\tau_t} + \boldsymbol{\theta}_{\tau_t} \cdot \mathbf{y}_t)}{\sum_{\tau' \in \mathcal{A}_t} \exp(q_{\tau'} + \boldsymbol{\theta}_{\tau'} \cdot \mathbf{y}_t)} \qquad (6)$$

with respect to the classifier and LSTM parameters. Note that the loss is differentiable with respect to the parameters; gradients are calculated

Figure 4: Example of a joint parse tree fragment with vector representations shown at each node. The vectors are obtained by recursive composition of representations of head, dependent, and label vectors. Syntactic dependencies and labels are in green, semantic in blue.

using backpropagation. We apply stochastic gradient descent with dropout for all neural network parameters.

### 3.5 Pretrained Embeddings

Following Dyer et al. (2015), "structured skip-gram" embeddings (Ling et al., 2015) were used, trained on the English (AFP section), German, Spanish and Chinese Gigaword corpora, with a window of size 5; training was stopped after 5 epochs. For out-of-vocabulary words, a randomly initialized vector of the same dimension was used.

### 3.6 Predicate Sense Disambiguation

Predicate sense disambiguation is handled within the model (M-PRED transitions), but since senses are lexeme-specific, we need a way to handle unseen predicates at test time. When a predicate is encountered at test time that was not observed in training, our system constructs a predicate from the predicted lemma of the word at that position and defaults to the "01" sense, which is correct for 91.22% of predicates by type in the English CoNLL 2009 training data.

## 4 Experimental Setup

Our model is evaluated on the CoNLL shared tasks on joint syntactic and semantic dependency parsing in 2008 (Surdeanu et al., 2008) and 2009 (Hajič et al., 2009). The standard training, development and test splits of all datasets were

used. Per the shared task guidelines, automatically predicted POS tags and lemmas provided in the datasets were used for all experiments. As a preprocessing step, pseudo-projectivization of the syntactic trees (Nivre et al., 2007) was used, which allowed an accurate conversion of even the non-projective syntactic trees into syntactic transitions. However, the oracle conversion of semantic parses into transitions is not perfect despite using the M-SWAP action due to the presence of multiple crossing arcs.[5]

The standard evaluation metrics include the syntactic labeled attachment score (LAS), the semantic $F_1$ score on both in-domain (WSJ) and out-of-domain (Brown corpus) data, and their macro average (Macro $F_1$) to score joint systems. Because the task was defined somewhat differently in each year, each dataset is considered in turn.

### 4.1 CoNLL 2008

The CoNLL 2008 dataset contains annotations from the Penn Treebank (Marcus et al., 1993), PropBank (Palmer et al., 2005) and Nom-Bank (Meyers et al., 2004). The shared task evaluated systems on predicate identification in addition to predicate sense disambiguation and SRL.

To identify predicates, we trained a zero-Markov order bidirectional LSTM two-class classifier. As input to the classifier, we use learned representations of word lemmas and POS tags. This model achieves an $F_1$ score of 91.43% on marking words as predicates (or not).

**Hyperparameters** The input representation for a word consists of pretrained embeddings (size 100 for English, 80 for Chinese, 64 for German and Spanish), concatenated with additional learned word and POS tag embeddings (size 32 and 10, resp.). Learned embeddings for syntactic and semantic arc labels are of size 20 and predicates 100. Two-layer LSTMs, with hidden state dimension 100 were used for each of the four stacks. The parser state, $\mathbf{y_t}$ and the composition function, $\mathbf{g}$ are of dimension 100. A dropout rate of 0.2 (Zaremba et al., 2014) was used on all layers at training time, tuned on the dev data from the set of values $\{0.1, 0.2, 0.3, 1.0\}$. The learned representations for actions are of size 100, similarly tuned from $\{10, 20, 30, 40, 100\}$. Other hyperpa-

---

[5]For 1.5% of English sentences in the CoNLL 2009 English dataset, the transition sequence incorrectly encodes the gold-standard joint parse; details in Henderson et al. (2013).

rameters have been set intuitively; careful tuning is expected to yield improvements (Weiss et al., 2015).

An initial learning rate of 0.1 for stochastic gradient descent was used and updated in every training epoch with a decay rate of 0.1 (Dyer et al., 2015). Training is stopped when the dev performance does not improve for approximately 6–7 hours of elapsed time. Experiments were run on a single thread on a CPU, with memory requirements of up to 512 MB.

### 4.2 CoNLL 2009

Relative to the CoNLL 2008 task (above), the main change in 2009 is that predicates are pre-identified, and systems are only evaluated on predicate sense disambiguation (not identification). Hence, the bidirectional LSTM classifier is not used here. The preprocessing for projectivity, and the hyperparameter selection is the same as in §4.1.

In addition to the joint approach described in the preceding sections, we experiment here with several variants:

**Semantics-only:** all syntactic transitions in $\mathcal{S}$, the syntactic stack $S$, and the syntactic composition function, $g_s$ are discarded. As a result, the set of constraints on transitions is a subset of the full set of constraints in §2.3. Effectively, this model does not use any syntactic features, similar to Collobert et al. (2011) and Zhou and Xu (2015). It provides a controlled test of the benefit of jointly predicting syntax in a semantic parser.

**Syntax-only:** all semantic transitions in $\mathcal{M}$, the semantic stack $M$, and the semantic composition function $g_m$ are discarded. S-SHIFT and S-RIGHT now *move* the item from the front of the buffer to the syntactic stack, instead of copying. The set of constraints on the transitions is again a subset of the full set of constraints. This model is an arc-eager variant of Dyer et al. (2015), and serves to check whether semantic parsing degrades syntactic performance.

**Hybrid:** the semantics parameters are trained using automatically predicted syntax from the syntax-only model. At test time, only semantic parses are predicted. This setup bears similarity to other approaches which pipeline syntax and semantics, extracting features from the syntactic parse to help SRL. However, unlike other

| Model | LAS | Sem. $F_1$ | Macro $F_1$ |
|---|---|---|---|
| *joint models:* | | | |
| Lluís and Màrquez (2008) | 85.8 | 70.3 | 78.1 |
| Henderson et al. (2008) | 87.6 | 73.1 | 80.5 |
| Johansson (2009) | 86.6 | 77.1 | 81.8 |
| Titov et al. (2009) | 87.5 | 76.1 | 81.8 |
| *CoNLL 2008 best:* | | | |
| #3: Zhao and Kit (2008) | 87.7 | 76.7 | 82.2 |
| #2: Che et al. (2008) | 86.7 | 78.5 | 82.7 |
| #2: Ciaramita et al. (2008) | 87.4 | 78.0 | 82.7 |
| #1: J&N (2008) | 89.3 | 81.6 | 85.5 |
| Joint (this work) | 88.9 | 80.2 | 84.6 |

Table 2: Joint parsers: comparison on the CoNLL 2008 test (WSJ+Brown) set.

approaches, this model does not offer the entire syntactic tree for feature extraction, since only the partial syntactic structures present on the syntactic stack (and potentially the buffer) are visible at a given timestep.

## 5 Results and Discussion

**CoNLL 2008 (Table 2)** Our joint model significantly outperforms the joint model of Henderson et al. (2008), from which our set of transitions is derived, showing the benefit of learning a representation for the entire algorithmic state. Several other joint learning models have been proposed (Lluís and Màrquez, 2008; Johansson, 2009; Titov et al., 2009) for the same task; our joint model surpasses the performance of all these models. The best reported systems on the CoNLL 2008 task are due to Johansson and Nugues (2008), Che et al. (2008), Ciaramita et al. (2008) and Zhao and Kit (2008), all of which pipeline syntax and semantics; our system's semantic and overall performance is comparable to these. We fall behind only Johansson and Nugues (2008), whose success was attributed to carefully designed global SRL features integrated into a pipeline of classifiers, making them asymptotically slower.

**CoNLL 2009 English (Table 3)** All of our models (Syntax-only, Semantics-only, Hybrid and Joint) improve over Gesmundo et al. (2009) and Henderson et al. (2013); demonstrating the benefit of our entire-parser-state representation learner compared to the more locally scoped model.

Given that syntax has consistently proven useful

in SRL, our Semantics-only model underperforms Hybrid and Joint, as expected. In the training domain, syntax and semantics benefit each other (Joint outperforms Hybrid). Out-of-domain (the Brown test set), the Hybrid pulls ahead, a sign that Joint overfits to WSJ. As a syntactic parser, our Syntax-only model performs slightly better than Dyer et al. (2015), who achieve 89.56 LAS on this task.

The overall performance of Joint is on par with the other winning participants at the CoNLL 2009 shared task (Zhao et al., 2009; Che et al., 2009; Gesmundo et al., 2009), falling behind only Zhao et al. (2009), who also carefully design language-specific features, and use a series of pipelines for the joint task, resulting in an accurate but computationally expensive system.

State-of-the-art SRL systems (shown in the last block of Table 3) which use advancements orthogonal to the contributions in this paper, perform better than our models. Many of these systems use expert-crafted features derived from full syntactic parses in a pipeline of classifiers followed by a global reranker (Björkelund et al., 2009; Björkelund et al., 2010; Roth and Woodsend, 2014); we have not used these features or reranking. Lei et al. (2015) use syntactic parses to obtain interaction features between predicates and their arguments and then compress feature representations using low-rank tensor. Täckström et al. (2015) present an exact inference algorithm for SRL based on dynamic programming and their local and structured models make use of many syntactic features from a pipeline; our search procedure is greedy. Their algorithm is adopted by FitzGerald et al. (2015) for inference in a model that jointly learns representations from a combination of PropBank and FrameNet annotations; we have not experimented with extra annotations.

Our system achieves an end-to-end runtime of 177.6±18 seconds to parse the CoNLL 2009 English test set on a single core. This is almost 2.5 times faster than the pipeline model of Lei et al. (2015) (439.9±42 seconds) under identical settings.[6]

**CoNLL 2009 Multilingual (Table 4)** We tested the joint model on the non-English CoNLL 2009 datasets, and the results demonstrate that it adapts

---

[6] See `https://github.com/taolei87/SRLParser`; we chose this system since it is publicly available, and other state-of-the-art systems are not.

| Model | LAS | Sem. $F_1$ (WSJ) | Sem. $F_1$ (Brown) | Macro $F_1$ |
|---|---|---|---|---|
| *CoNLL'09 best:* | | | | |
| #3 G+ '09 | 88.79 | 83.24 | 70.65 | 86.03 |
| #2 C+ '09 | 88.48 | 85.51 | 73.82 | 87.00 |
| #1 Z+ '09a | 89.19 | 86.15 | 74.58 | 87.69 |
| *this work:* | | | | |
| Syntax-only | 89.83 | | | |
| Sem.-only | | 84.21 | 73.72 | |
| Hybrid | 89.83 | 84.58 | 75.64 | 87.20 |
| Joint | 89.94 | 84.97 | 74.48 | 87.45 |
| *pipelines:* | | | | |
| R&W '14 | | 86.34 | 75.90 | |
| L+ '15 | | 86.58 | 75.57 | |
| T+ '15 | | 87.30 | 75.50 | |
| F+ '15 | | 87.80 | 75.50 | |

Table 3: Comparison on the CoNLL 2009 English test set. The first block presents results of other models evaluated for both syntax and semantics on the CoNLL 2009 task. The second block presents our models. The third block presents the best published models, each using its own syntactic pre-processing.

easily—it is on par with the top three systems in most cases. We note that our Chinese parser relies on pretrained word embeddings for its superior performance; without them (not shown), it was on par with the others. Japanese is a small-data case (4,393 training examples), illustrating our model's dependence on reasonably large training datasets.

We have not extended our model to incorporate morphological features, used by the systems to which we compare. Future work might incorporate morphological features where available; this could potentially improve performance, especially in highly inflective languages like Czech. An alternative might be to infer word-internal representations using character-based word embeddings, which was found beneficial for syntactic parsing (Ballesteros et al., 2015).

| Language | #1 C+'09 | #2 Z+'09a | #3 G+'09 | Joint |
|---|---|---|---|---|
| Catalan | 81.84 | **83.01** | 82.66 | 82.40 |
| Chinese | 76.38 | 76.23 | 76.15 | **79.27** |
| Czech | **83.27** | 80.87 | 83.21 | 79.53 |
| English | 87.00 | **87.69** | 86.03 | 87.45 |
| German | **82.44** | 81.22 | 79.59 | 81.05 |
| Japanese | **85.65** | 85.28 | 84.91 | 80.89 |
| Spanish | 81.90 | **83.31** | 82.43 | 83.11 |
| Average | 82.64 | 82.52 | 82.14 | 81.96 |

Table 4: Comparison of macro $F_1$ scores on the multilingual CoNLL 2009 test set.

## 6 Related Work

Other approaches to joint modeling, not considered in our experiments, are notable. Lluís et al. (2013) propose a graph-based joint model using dual decomposition for agreement between syntax and semantics, but do not achieve competitive performance on the CoNLL 2009 task. Lewis et al. (2015) proposed an efficient joint model for CCG syntax and SRL, which performs better than a pipelined model. However, their training necessitates CCG annotation, ours does not. Moreover, their evaluation metric rewards semantic dependencies regardless of where they attach within the argument span given by a PropBank constituent, making direct comparison to our evaluation is infeasible. Krishnamurthy and Mitchell (2014) propose a joint CCG parsing and relation extraction model which improves over pipelines, but their task is different from ours. Li et al. (2010) also perform joint syntactic and semantic dependency parsing for Chinese, but do not report results on the CoNLL 2009 dataset.

There has also been an increased interest in models which use neural networks for SRL. Collobert et al. (2011) proposed models which perform many NLP tasks without hand-crafted features. Though they did not achieve the best results on the constituent-based SRL task (Carreras and Màrquez, 2005), their approach inspired Zhou and Xu (2015), who achieved state-of-the-art results using deep bidirectional LSTMs.[7] Our approach for dependency-based SRL is not directly comparable.

## 7 Conclusion

We presented an incremental, greedy parser for joint syntactic and semantic dependency parsing. Our model surpasses the performance of previous joint models on the CoNLL 2008 and 2009 English tasks, despite not using expert-crafted features of the full syntactic parse, as done by asymptotically more expensive models. An open-source implementation of our parser will be made available on publication.

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
