# Peer review of "Greedy, Joint Syntactic-Semantic Parsing with Stack LSTMs"

_CoNLL 2016 — decision unknown_

[Official Review · Reviewer 1 · rating 5 · confidence 5]
soundness 5 · originality 3 · clarity 5 · impact 4 · substance 5 · appropriateness 5 · meaningful comparison 5 · replicability 4 · presentation format Oral Presentation

This paper presents a Stack LSTM parser based on the work of Henderson et al.
(2008, 2013) on joint syntactic/semantic transition-based parsing and Dyer et
al. (2015) on stack LSTM syntactic parsing. The use of the transition system
from the former and the stack LSTM from the latter shows interesting results
compared to the joint systems on the CoNLL 2008 and 2009 shared tasks.

I like this paper a lot because it is well-written, well-explained, the related
work is good and the results are very interesting. The methodology is sound
(with a minor concern regarding the Chinese embeddings, leading me to believe
than very good embeddings can be more informative than a very clever model...).

Moreover, the description of the system is clear, the hyperparameters are
justified and the discussion is interesting.

The only thing I would say is that the proposed system lacks originality in the
sense that the work of Henderson et al. puts the basis of semi-synchronised
joint syntax-semantic transition-based parsing several years ago and Dyer et
al. came up with the stack LSTM last year, so it is not a new method, per say.
But in my opinion, we were waiting for such a parser to be designed and so I'm
glad it was done here.

[Official Review · Reviewer 2 · rating 4 · confidence 3]
soundness 4 · originality 3 · clarity 4 · impact 3 · substance 4 · appropriateness 5 · meaningful comparison 4 · replicability 4 · presentation format Oral Presentation

General comments
================

The paper presents a joint syntactic and semantic transition-based dependency
parser,
inspired from the joint parser of Henderson et al. (2008).
The authors claim two main differences:
- vectorial representations are used for the whole parser's state, instead of
the top elements of the stack / the last parser's configurations
- the algorithm is a plain greedy search

The key idea is to take advantage of stack LSTMs so that the vector
representing the state of the parser
keeps memory of potentially large scoped syntactic features, which
are known to be decisive features for semantic role labeling
(such as the path between the predicate and the candidate role filler head).

The system is tested on the CoNLL 2008 data set (English) and on the
multilingual CoNLL 2009 data set.
The authors compare their system's performance to previously reported
performances,
showing their system does well compared to the 2008 / 2009 systems, 
but less compared to more recent proposals (cf. bottom of table 3).
They emphasized though that the proposed system does not require any hand-craft
features,
and is fast due to the simple greedy algorithm.

The paper is well written and describes a substantial amount of work,
building on the recently popular LSTMs, applied to the Henderson et al.
algorithm
which appears now to have been somewhat visionary.

I have reservations concerning the choice of the simple greedy algorithm:
it renders results not comparable to some of the cited works.
It would not have been too much additional work nor space to provide for
instance beam-searched performance.

More detailed comments / questions
==================================

Section 2:

A comment on the presence of both A1 and C-A1 links would help understanding
better the target task of the paper.

A summary of the differences between the set of transitions used in this work
and that of Henderson et al. should be provided. In its current form, it is
difficult to 
tell what is directly reused from Henderson et al. and what is new / slightly
modified.

Section 3.3

Why do you need representations concatenating the word predicate and its
disambiguated sense,
this seems redundant since the disambiguated sense are specific to a predicate
?

Section 4

The organization if the 4.1 / 4.2 sections is confusing concerning
multilinguality.
Conll 2008 focused on English, and CoNLL 2009 shared task extended it to a few
other languages.

[Official Review · Reviewer 3 · rating 5 · confidence 4]
soundness 5 · originality 3 · clarity 5 · impact 4 · substance 4 · appropriateness 5 · meaningful comparison 5 · replicability 5 · presentation format Oral Presentation

This paper performs an overdue circling-back to the problem of joint semantic
and syntactic dependency parsing, applying the recent insights from neural
network models. Joint models are one of the most promising things about the
success of transition-based neural network parsers.

There are two contributions here. First, the authors present a new transition
system, that seems better than the Hendersen (2008) system it is based on. The
other contribution is to show that the neural network succeeds on this problem,
where linear models had previously struggled. The authors attribute this
success to the ability of the neural network to automatically learn which
features to extract. However, I think there's another advantage to the neural
network here, that might be worth mentioning. In a linear model, you need to
learn a weight for each feature/class pair. This means that if you jointly
learn two problems, you have to learn many more parameters. The neural network
is much more economical in this respect.

I suspect the transition-system would work just as well with a variety of other
neural network models, e.g. the global beam-search model of Andor (2016). There
are many other orthogonal improvements that could be made. I expect extensions
to the authors' method to produce state-of-the-art results.

It would be nice to see an attempt to derive a dynamic
oracle for this transition system, even if it's only in an appendix or in
follow-up work. At first glance, it seems similar to the
arc-eager oracle. The M-S action excludes all semantic arcs between the word at
the start of the buffer and the words on the semantic stack, and the M-D action
excludes all semantic arcs between the word at the top of the stack and the
words in the buffer. The L and R actions seem to each exclude the reverse arc,
and no other.